

# Comparison of lipid accumulation product and body mass index as indicators of diabetes diagnosis among 215,651 Chinese adults

Tian Tian[1], Hualian Pei[1], Zhen Chen[1], Gulisiya Hailili[1], Shuxia Wang[2], Yong Sun[2], Hua Yao[2] and Dai Jianghong[1]

[1] School of Public Health, Xinjiang Medical University, Urumqi, China
[2] School of Health Management, Xinjiang Medical University, Urumqi, China

## ABSTRACT

**Purpose:** We aimed to assess if lipid accumulation product (LAP) could outperform body mass index (BMI) as a marker for diabetes diagnosis.

**Methods:** We analyzed the results of a national physical examination project in Urumqi, China. This project was conducted in 442 community clinics in Urumqi from October 2016 to February 2017.

**Results:** LAP was highly correlated with diabetes. The subjects with higher amounts of LAP had a higher risk of diabetes, and the prevalence of diabetes in the fourth quartile of LAP was dramatically higher than in the first quartile (5.72% vs. 21.76%). The adjusted odds ratios (AOR) associated with diabetes in the fourth quartile of LAP was significantly higher than the AOR associated with diabetes in the first quartile, and when BMI $\geq$ 28 kg/m$^2$ was compared with BMI < 28 kg/m$^2$ (3.24 (3.11, 3.37) vs. 1.65 (1.60, 1.70)). The LAP's area under the curve (AUC) was significantly higher than the BMI's AUC when based on diabetes (0.655 vs. 0.604). In the normal BMI group, 34% of participants had a LAP value higher than the cutoff point found during ROC analysis. In this subgroup, we observed a significantly higher prevalence of diabetes that was similar to that of the subgroup with a BMI $\geq$ 28 kg/m$^2$, and both of their LAP values were higher than the cutoff point.

**Conclusion:** When use as a tool for diabetes diagnosis, LAP performed better than BMI, implying that LAP could be a preferable anthropometry assessment.

## INTRODUCTION

Over the past years, the prevalence of type 2 diabetes has increased considerably worldwide, with more notable increases in developing countries (*IDF Clinical Guidelines Task Force, 2006*). China, the largest middle-income country, has the largest diabetes epidemic in the world, with a prevalence of 10.9% in 2013 (*Bozorgmanesh, Hadaegh & Azizi, 2010*). Diabetes is linked with a number of vascular and nonvascular complications (*Browning, Hsieh & Ashwell, 2010*).

Corresponding authors
Hua Yao, yaohua01@sina.com
Dai Jianghong, epidjh@163.com

The pathogen of diabetes is not well defined, but inducing factors include genetics, inflammation, aging, lifestyle, and obesity (*Despres, 2006*). The risk of diabetes in adults increases with increasing adiposity, and decreases with fat loss. Adiposity leads to insulin resistance, is connected to molecules that make individuals susceptible to inflammation and metabolic complications (*Gao et al., 2013*), and is believed to be a promoter of type 2 diabetes mellitus. There are a number of techniques designed to estimate obesity, such as underwater weighing, dual-energy X-ray absorption, computed tomography scans and magnetic resonance imaging. These measurements have their own advantages—while some are most precise for measuring body fat, others can evaluate body fat distribution (*Janssen, Katzmarzyk & Ross, 2004*; *Kahn, 2005*; *Kishida et al., 2012*; *Lee et al., 2008*). However, these technologically complicated approaches are too expensive and time-consuming to be applied in large-scale investigations.

In the past few years, anthropometric indicator measurements such as body mass index (BMI) have been recognized as cost-effective ways to evaluate obesity. BMI typically represents overall obesity, while other measurements are applied to indicate central obesity. However, BMI is now being cautioned as an inadequate indicator of individual obesity, and an increasing number of studies have suggested that central adiposity may be a better indicator of diabetes mellitus (*Browning, Hsieh & Ashwell, 2010*; *Lee et al., 2008*; *Park et al., 2009*; *Taylor et al., 2010*; *Tseng et al., 2010*). For central adiposity, waist circumference (WC) has been considered as a lucid estimation. *Motamed et al. (2016)* found a notable association between triglyceride (TG) level and risk of metabolic syndrome. Furthermore, there is proof that increased WC is associated with the aggregation of TG levels. A number of studies have shown that contemporaneous quantification of TG levels and WC could indicate a symptom of lipid overaccumulation related to the risk of metabolic syndrome. *Kahn (2005)* described a simple index, counting as the compound of WC and fasting plasma TG levels, in order to estimate excessive lipid accumulation in adults. He utilized the lipid overaccumulation concept, and tested the assumption that the mentioned index is connected with various cardiovascular risk factors more effectively than BMI.

An increasing number of studies view LAP as an effective marker for diabetes (*Bozorgmanesh, Hadaegh & Azizi, 2010*), metabolic syndrome (*Motamed et al., 2016*; *Taverna et al., 2011*), and insulin resistance (*Xia et al., 2012*) among the overall population. These findings inspired us to test the relationship between LAP and diabetes in a large-scale population. Our major point in this research was to evaluate whether LAP is better than BMI as an indicator of diabetes among the overall population, and to find out its optimal cutoff point for diabetes diagnosis.

## MATERIALS AND METHODS

### Population

Beginning in September 2016, Xinjiang, the largest autonomous region in China, launched a comprehensive physical examination project within the region. All residents in Xinjiang can participate in this free annual physical examination, with the cost covered by the local government. The residents were recruited by government announcement. Urumqi,
the capital of Xinjiang, is the center of Silk Road economic zone, and understanding the health status of the citizens of Urumqi is one of the most important purposes of the national physical examination project. From October 2016 to February 2017, 303,620 subjects in Urumqi participated in this physical examination project. According to the sixth census and the projection of the annual report of the Public Security Bureau, there were 3.12 million permanent residents in Urumqi in 2016. The Medical Ethics Committee of the Fourth Affiliated Hospital of Xinjiang Medical University approved the study protocol (2018XE0108). Informed written consent was obtained from all subjects.

The present research examined data from Urumqi. Participants over 18 years qualified for the current analysis. Participants who had incomplete information on variables (WC, fasting plasma TG, height, weight) used to calculate LAP and BMI were excluded. The final sample of this analysis consisted of 215,651 participants.

## Data collection and laboratory tests

Four hundred and forty-two community clinics in Urumqi were involved in this national physical examination project. All community clinic staff involved in this project were trained on the protocol established by the National Health Commission of Xinjiang. The training included questionnaires, physical measurements, laboratory tests, and data entry. Participants were interviewed in the community health center closest to their house, and the interview location for each participant was recorded. Trained interviewers use pre-tested questionnaires to gather information. The collected information included demographic data and behavior risk factors.

Weight was valued using digital scales with participants wearing as few clothes as possible without shoes, and the weight reading was accurate to 100 g. Height was valued by a tape meter in the standing position, without shoes and with the shoulders aligned normally. When measuring the WC, an unstretched tape meter was kept at the umbilical level and recorded to the nearest 0.1 cm, making sure that there was no pressure on the body surface. All the participants wore as few clothes as possible.

After the subjects were fasted for at least 8 h, the investigators collected five mL of blood samples from the subjects and placed blood samples in vacutainer tubes of ethylenediaminetetraacetic acid (EDTA) for determination of glucose and other indices. Serum concentrations of TG, fasting glucose, serum total cholesterol and high-density lipoprotein (HDL) were tested through a biochemical analyzer (Dimension AR/AVL Clinical Chemistry System, Newark, NJ, USA) in the Laboratory of the community clinic where the participant had been interviewed.

## Definition of variables and outcomes

Body mass index (kg/m$^2$) is equal to body weight (kg) divided by the square of the height (m). According to the Chinese definitions, participants whose BMI $\geq$ 28 kg/m$^2$ were defined as obese (*Zhou, 2002*). For men, LAP = (WC (cm) − 65) × (TG (mmol/L)); for women, LAP = (WC (cm) − 58) × (TG (mmol/L)). A diabetes diagnosis was defined as having a history of diabetes mellitus, if they were using antidiabetic agents currently, or a FPG level $\geq$ 7.0 mmol/l (2006).

## Statistical analysis

The categorical variables of participants' demographic characteristics were presented in numbers and percentage. Continuous variables were displayed in mean ± standard deviation (SD). For categorical variables, we used Pearson's $\chi^2$ test to compare basic characteristics of different diabetes statuses (with/without), and for continuous variables, we used Student's $t$-tests. In order to get a profound understanding of the connection between LAP levels and the epidemic of diabetes, we split the study participants into four groups according to LAP quartiles. We used Pearson's $\chi^2$ test to compare the discrepancy of diabetes prevalence in different groups. To calculate the odds ratios (ORs) with 95% confidence intervals (CIs) and adjusted odds ratios (AORs) with 95% CIs, we used logistic regression, and then the ORs and AORs were used to estimate the risk of diabetes in every LAP quartile, setting the lowest quartile of LAP as reference. Age, gender, education, smoking and alcohol consumption were all entered into the multivariate logistic model to calculate the AOR. We calculated the area under the curve (AUC) using receiver operating characteristics (ROC), and the AUC was used to estimate the intensity of the relationship between subdivided LAP level and diabetes. At the highest Youden Index ((specificity + sensibility) − 1), the best cutoff point was determined, and the discrimination among different AUCs was tested by a nonparametric method. We used R 3.4.1 for all the analyses and $P$ values of <0.05 (two-tailed) were considered as statistically significant.

# RESULTS

## Characteristics of the study subjects

During the study, 215,651 Chinese subjects were included. The characteristics of all participants, with and without diabetes, are outlined in Table 1. The mean age of all subjects was 50.02 years old. Of them, 55.86% were female, 34.71% had received a junior middle school education, 16.72% were current smokers and 16.09% were current drinkers. Among the 215,651 subjects, 27,917 had diabetes, and the prevalence of diabetes was 12.95%. Patients with diabetes were older and had higher weights and TG levels than those subjects without diabetes (Table 1).

## Association between lipid accumulation product and diabetes

To more efficiently uncover the relationship between LAP and the prevalence of diabetes, the participants were divided into four subgroups based on LAP level. According to the survey, the prevalence of diabetes was greater in the higher LAP quartile groups. We also divided the subjects according to their BMI, and the prevalence of diabetes was expressively higher in the subjects who had higher BMI (Table 2).

The non-adjusted and adjusted analysis association among quartiles of LAP/BMI are shown in Table 3. In the multiple logistic regression model, confounding factor including age, education, smoking and alcohol consumption were entered into the model to calculate the AOR, and we did not eliminate any variables. Compared to the first quartile of LAP, the COR associated with diabetes of the second (OR 95% CI 1.28 [1.22–1.34]), third (OR 95% CI 1.86 [1.78–1.95]) and fourth quartile (OR 95% CI 4.67 [4.49–4.86]) of LAP were significant; the COR associated with diabetes for a BMI ≥ 28 kg/m² (OR 95% CI 1.65

**Table 1 Characteristics of the study subjects with and without diabetes.**

| Variables | With diabetes N (%) | Without diabetes N (%) | P value |
|---|---|---|---|
| Age* (year, $\bar{x} \pm s$) | 58.14 ± 12.93 | 48.81 ± 14.83 | <0.001 |
| Gender | | | <0.001 |
| Male | 13,472 (14.1) | 81,722 (85.9) | |
| Female | 14,445 (12.0) | 106,012 (88.0) | |
| Education# | | | <0.001 |
| Illiteracy | 1,902 (19.3) | 7,967 (80.7) | |
| Primary school | 7,064 (15.3) | 39,225 (84.7) | |
| Junior high school | 9,746 (13.0) | 65,122 (87.0) | |
| Senior high school | 5,519 (12.2) | 39,784 (87.8) | |
| University or higher | 2,922 (8.7) | 27,580 (91.3) | |
| Unspecified | 764 (8.5) | 8,045 (91.5) | |
| Height (cm, $\bar{x} \pm s$) | 163.3 ± 8.6 | 164.0 ± 8.2 | <0.001 |
| Weight (kg, $\bar{x} \pm s$) | 69.9 ± 11.8 | 67.0 ± 11.9 | <0.001 |
| WC (cm, $\bar{x} \pm s$) | 90.6 ± 10.5 | 85.8 ± 10.8 | <0.001 |
| FPG (mmol/L, $\bar{x} \pm s$) | 7.74 ± 2.80 | 5.10 ± 0.90 | <0.001 |
| TG (mmol/L, $\bar{x} \pm s$) | 2.07 ± 1.66 | 1.57 ± 1.15 | <0.001 |
| HDL-C (mmol/L, $\bar{x} \pm s$) | 1.50 ± 0.62 | 1.49 ± 0.56 | <0.001 |
| Smoking status& | | | <0.001 |
| Non-smokers | 21,254 (12.3) | 151,163 (87.7) | |
| Previous smokers | 1,483 (21.4) | 5,438 (78.6) | |
| Current smokers | 5,128 (14.2) | 30,919 (85.8) | |
| Alcohol consumption^ | | | <0.001 |
| No | 22,564 (12.5) | 158,173 (87.5) | |
| Yes | 5,305 (15.3) | 29,401 (84.7) | |

Notes:
There were missing values on participants' age, education, smoking status, alcohol consumption, and the number of missing were 353, 11, 266, 208, respectively.
WC, waist circumference; FBG, fasting blood glucose; TG, triglyceride; HDL-C, high-density lipoprotein cholesterol.

[1.63–1.74]) was significant when compared with a BMI < 28 kg/m². Compared with the first quartile, the risks associated with diabetes for the third and last quartiles of LAP were both still significant after adjusted age, gender, education, smoking status and alcohol consumption. The AORs 95% CI were 0.97 [0.92–1.02], 1.28 [1.23–1.34] and 3.24 [3.11–3.37] for the second, third and last quartiles of LAP, respectively. Meanwhile, the AOR associated with diabetes for a BMI ≥ 28 kg/m² (OR 95% CI 1.65 [1.60–1.70]) was also still significant when compared with a BMI < 28 kg/m² (Table 3).

## Evaluation of the predictive accuracy of LAP and BMI

LAP exhibited higher diagnostic accuracy for diabetes compared to BMI (P < 0.0001). For LAP, the optimum cutoff points were 38.045 (sensitivity = 60.6%, specificity = 62.0 %), while for BMI, the optimum cutoff points were 25.065 (sensitivity = 55.2%, specificity = 60.2%) (Table 4).

**Table 2 Distribution of LAP and BMI among subjects with and without diabetes.**

| Variables | With diabetes N (%) | Without diabetes N (%) | P value |
|---|---|---|---|
| Quartiles of LAP | | | <0.001 |
| 1st (≤18.64) | 3,127 (5.7) | 51,509 (94.3) | |
| 2nd (18.65~32.95) | 5,291 (10.0) | 47,869 (90.0) | |
| 3rd (32.96~55.67) | 7,928 (14.7) | 46,007 (85.3) | |
| 4th (≥55.68) | 11,571 (21.5) | 42,349 (78.5) | |
| BMI (kg/m$^2$) | | | |
| <28 | 20,229 (11.7) | 152,470 (88.3) | <0.001 |
| ≥28 | 7,688 (17.9) | 35,264 (82.1) | |

**Table 3 Odds ratios and 95% confidence intervals for diabetes according to quartiles of LAP and BMI.**

| Variables | COR (95% CI) | P value | AOR (95% CI) | P value |
|---|---|---|---|---|
| Quartiles of LAP | | | | |
| 1st (≤18.64) | | | | |
| 2nd (18.65~32.95) | 1.28 [1.22–1.34] | <0.001 | 0.97 [0.92–1.02] | 0.24 |
| 3rd (32.96~55.67) | 1.86 [1.78–1.95] | <0.001 | 1.28 [1.23–1.34] | <0.001 |
| 4th (≥55.68) | 4.67 [4.49–4.86] | <0.001 | 3.24 [3.11–3.37] | <0.001 |
| BMI (kg/m$^2$) | | | | |
| <28 | | | | |
| ≥28 | 1.65 [1.63–1.74] | <0.001 | 1.65 [1.60–1.70] | <0.001 |

Note:
COR, crude odds ratio; AOR, adjusted odds ratio; 95% CI, 95% confidence interval. Age, gender, education, smoking and alcohol consumption were adjusted for calculated AOR.

**Table 4 Assessment of the predictive accuracy of LAP and BMI by gender.**

| Variables | | AUC (95% CI) | Cut-off | Sensitivity (%) | Specificity (%) |
|---|---|---|---|---|---|
| Total | LAP | 0.655 [0.652–0.658] | 38.41 | 60.6 | 62.0 |
| | BMI | 0.604 [0.600–0.607] | 25.07 | 60.2 | 55.2 |
| Male | LAP | 0.625 [0.621–0.630] | 35.71 | 64.5 | 53.3 |
| | BMI | 0.580 [0.576–0.586] | 24.84 | 66.7 | 45.8 |
| Female | LAP | 0.679 [0.674–0.684] | 33.05 | 71.1 | 56.1 |
| | BMI | 0.618 [0.614–0.623] | 24.23 | 65.8 | 51.9 |

Note:
AUC, area under the curve.

## Distribution of LAP according to BMI

After divided according to BMI (<28 kg/m$^2$ or ≥28 kg/m$^2$), the participants were subdivided into two groups according to the cutoff point of LAP obtained in this study, and the distribution of LAP values in subjects with and without diabetes were compared. In the normal BMI group, 34% of subjects had a LAP higher than the cutoff point, and their prevalence of diabetes was significantly higher than the group whose LAP was

**Table 5 Distribution of LAP according to gender and BMI.**

| Group | | LAP | With diabetes N (%) | Without diabetes N (%) | P value |
|---|---|---|---|---|---|
| Gender* | Male | <38.045 | 5,238 (10.2) | 46,140 (89.8) | <0.001 |
| | | ≥38.045 | 8,234 (18.8) | 35,582 (81.2) | |
| | Female | <38.045 | 5,226 (7.3) | 66,626 (92.7) | <0.001 |
| | | ≥38.045 | 9,219 (19.0) | 39,386 (81.0) | |
| BMI (kg/m$^2$)# | <28 | <38.045 | 9,409 (8.3) | 104,497 (91.7) | <0.001 |
| | | ≥38.045 | 10,820 (18.4) | 47,973 (81.6) | |
| | ≥28 | <38.045 | 1,055 (11.4) | 8,269 (88.7) | <0.001 |
| | | ≥38.045 | 6,633 (19.7) | 26,995 (80.3) | |

**Note:**
* The distribution of LAP between males and females was significantly different ($P < 0.001$).
# The distribution of LAP between BMI < 28 kg/m$^2$ and BMI ≥ 28 kg/m$^2$ was significantly different ($P < 0.001$).

below the cutoff value. We also compared the distribution of lap among males and females, the results indicated that there was significantly different ($P < 0.001$) (Table 5).

## DISCUSSION

Obesity has received extensive attention as a risk factor for diabetes in recent years, but there is still controversy about which obesity index performs better in a population study. LAP is uncomplicated, cheap, and could be useful when height and/or weight are difficult to assess (such as, when a person has lost a limb). In our study, we analyzed data from a national physical examination project, tried to determine the relationship between LAP and diabetes, and compared LAP with BMI for diabetes diagnostic accuracy. The results indicated that LAP strongly correlated with diabetes. The subjects with higher LAP had a higher risk of diabetes, the prevalence of diabetes in the fourth quartile of LAP was dramatically higher than in the first quartile (5.72% vs. 21.76%), and the conclusion was consistent with other studies. For example, *Shen et al. (2017)* conducted a cross-sectional study in Beijing, the capital of China, and concluded that an elevated level of LAP was linked to an increased risk of diabetes in adults, and that LAP could be an key predictor for diabetes. Wakabayashi I and his team carried out a number of studies to confirm the relationship between LAP and metabolic diseases such as diabetes and atherosclerosis. All his surveys showed a positive correlation, while some of his studies had taken the age and gender of subjects into consideration (*Wakabayashi, 2014*; *Wakabayashi & Daimon, 2012, 2014*).

The AOR associated with diabetes in the fourth quartile of LAP when compared with the first quartile was significantly higher than the AOR of BMI ≥ 28 kg/m$^2$ when compared with a BMI < 28 kg/m$^2$ (3.24 (3.11,3.37) vs. 1.65 (1.60,1.70)), and LAP's AUC based on diabetes was significantly higher than BMI's AUC based on diabetes (0.655 vs. 0.604). This shows that LAP has a stronger association with diabetes than BMI does. A number of studies have found similar results. An investigation conducted among Mongolians in China compared the relationship of LAP and BMI with the risk of hypertension, and they found LAP was better than BMI (*Gao et al., 2013*). Xia C. implied that LAP was a

more effective indicator than WC and BMI when identifying insulin resistance in non-diabetic individuals (*Gao et al., 2013*). Assessment of the predictive accuracy of LAP and BMI also confirmed that LAP is a better obesity index than BMI for the diagnosis of diabetes. This is not only supported by the data in our survey, but also by other research that has evaluated LAP and BMI's diagnostic sensitivity and specificity for metabolic disease (*Motamed et al., 2016*; *Taverna et al., 2011*).

BMI is treated as a scale of overweight and obesity, but should be used conservatively in epidemiological studies to estimate the health risk caused by the overaccumulation of body fat (*Kishida et al., 2012*). Subjects of different ages, sex and ethnicities can show high variability or may have the same BMI when we use BMI to interpret overweight and obese body composition. Overaccumulation of body fat in the viscera presents an additional health risk, but BMI does not imply any thorough understanding of partial body fat distribution and accumulation. In our study, after finding the optimal cutoff points, we found that in the normal BMI group, 34% of participants still had a LAP value higher than the cutoff point, and the prevalence of diabetes was significantly higher than in the other subgroups It was even close to the subgroup whose BMI $\geq$ 28 kg/m$^2$ and whose LAP value was higher than the cutoff point. Thus, BMI is not an accurate scale of body composition and distribution in individuals (*Prentice & Jebb, 2001*). WC is an important additional piece of information for assessing abdominal obesity that may better explain obesity-related health risk (*Janssen, Katzmarzyk & Ross, 2004*). It is easy to measure and shows high stability between different surveys (*Wang et al., 2003*). Although it is better to use WC to evaluate total abdominal fat, WC cannot make a distinction between visceral adiposity and subcutaneous abdominal fat, while fat accumulation in the viscera is highly correlated with metabolic abnormalities (*Despres, 2006*; *Kishida et al., 2012*). Serum TG level, even within the accepted normal range, is an independent risk factor for impaired fasting glucose and diabetes mellitus. LAP is an obesity index combining WC and TG together, so it is quite reasonable that it outperforms BMI in diagnosing diabetes and other metalogic diseases.

This large-scale project also allowed us to understand the overall prevalence and distribution of diabetes and other chronic diseases in Urumqi. The latest countrywide typical cross-sectional study in mainland China was conducted in 2013, which included 170,287 subjects, and concluded that the prevalence of diabetes was 10.9%. They estimated that the prevalence of diabetes in underdeveloped areas was 9.6%, although Urumqi, where our study conducted, is a typical underdeveloped city, and our result was significantly higher at 12.95%. We assumed that this discrepancy in diabetes prevalence was due to the participant criteria, as the previous study designed a multistage sampling method (*Wang et al., 2017*; *Xu et al., 2013*; *Zhou et al., 2015*), while we recruited volunteer subjects to join this project, implying that the participants in our study were more likely to be in poor physical conditions. This also implies that our research may overestimate the prevalence of diabetes in Urumqi, but the focus of our research is in the comparison of the diagnostic value of LAP and BMI of diabetes. This overestimation does not seem to have a substantive impact on our conclusion.

We should mention our study has several limitations. First of all, we did not have data to compare the participants to non-participants, but during the project, we found that most of the participants were not covered by other physical examination project, which means them were poorer than the non-participants. Second, as a cross-sectional study, and thus a causal association between LAP and diabetes cannot be determined. We encourage additional longitudinal studies to confirm our conclusions.

## CONCLUSIONS

In summary, this study indicated that LAP could be a better anthropometric measurement to diagnosis diabetes. In the future, prospective cohort studies should be conducted to test the causal association between LAP and diabetes.

### Funding

This work was supported by the national key research and development plan "precise medical research" key special sub-project "Xinjiang multi-ethnic natural population cohort construction and health follow-up study" (2017YFC0907203) and Xinjiang Uygur Autonomous Region "13th Five-Year" Key Discipline (Plateau discipline)—Public Health and Preventive Medicine. The funders had no role in study design, data collection and analysis, decision to publish, or preparation of the manuscript.

### Grant Disclosures

The following grant information was disclosed by the authors:
Xinjiang Multi-Ethnic Natural Population Cohort Construction and Health Follow-up Study: 2017YFC0907203.
Xinjiang Uygur Autonomous Region "13th Five-Year" Key Discipline (Plateau discipline)—Public Health and Preventive Medicine.

### Competing Interests

The authors declare that they have no competing interests.

### Author Contributions

- Tian Tian conceived and designed the experiments, performed the experiments, analyzed the data, prepared figures and/or tables, authored or reviewed drafts of the paper, and approved the final draft.
- Hualian Pei performed the experiments, analyzed the data, prepared figures and/or tables, and approved the final draft.
- Zhen Chen performed the experiments, prepared figures and/or tables, and approved the final draft.
- Gulisiya Hailili performed the experiments, prepared figures and/or tables, and approved the final draft.
- Shuxia Wang conceived and designed the experiments, prepared figures and/or tables, and approved the final draft.
- Yong Sun conceived and designed the experiments, performed the experiments, authored or reviewed drafts of the paper, and approved the final draft.
- Hua Yao performed the experiments, authored or reviewed drafts of the paper, and approved the final draft.
- Dai Jianghong conceived and designed the experiments, performed the experiments, authored or reviewed drafts of the paper, and approved the final draft.

### Human Ethics

The following information was supplied relating to ethical approvals (i.e., approving body and any reference numbers):

The Medical Ethics Committee of the Fourth Affiliated Hospital of Xinjiang Medical University approved the study protocol (2018XE0108).

### Data Availability

The raw data is available as a Supplemental File.

### Supplemental Information

Supplemental information for this article can be found online at http://dx.doi.org/10.7717/peerj.8483#supplemental-information.

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
