# Peer review of "Comparison of lipid accumulation product and body mass index as indicators of diabetes diagnosis among 215,651 Chinese adults"

_PeerJ, doi:10.7717/peerj.8483_

## Round 0.1 · original submission · Major Revisions

Dear authors,

Based on the comments of the reviewers, your manuscript needs some modifications to improve some issues highlighted in their reports. Please, see the comments below so as to have more information.

Best regards,
Dr Palazón-Bru (academic editor for PeerJ)

Reviewer 1 ·

Basic reporting

The manuscript needs proofreading. There are editing errors in the text and in the tables.

Experimental design

The BMI cut-off value for obesity used in this study needs to be described because it is the standard cut-off used in China but not necessary other parts of the world.

The multiple logistic regression analyses need to be described in more details.

Validity of the findings

The conclusion, the title, and the aim of the study did not match well.
The title suggested that the study was to compare LAP and BMI as indicators of diabetes risk. However, the purpose of the study appeared to "find the optimal cutoff point for diabetes diagnosis". Furthermore, in the conclusions, the authors stated that "LAP could be a better anthropometric measurement to predict the risk of diabetes". It is not clear whether the authors plan to use LAP simply as a tool for diabetes diagnosis or to predict risk of developing diabetes (which is not possible with a cross-sectional study design).

Additional comments

Title: The target population should be mentioned in the title as “….. diabetes risk in 215,651 Chinese adults”.

The authors need to carefully proofread the manuscript to correct the typographical errors, such as (1) The “N” after Motamed in Line 54 should be deleted; (2) “liquid” should be “lipid” (Line 60); (3) the (mean +/- S) for BMI should be deleted in Table 3; (4) The word “proved” should not be used (Line 62).

Methods: The use of 28 kg/m2 As the cutoff value for obesity needs to be explained that it is the standard point used in China. There are international variations in the cutoff value.

Methods: The variables included in the multiple logistic regression model should be shown along with the LAP and BMI so readers can get a sense of their magnitude and statistical significance. In addition, please mention whether procedures, such as stepwise, was used to eliminate variables that were not significantly associated with diabetes in the multiple logistic regression model.

Discussion: The authors mentioned that “we did not review the course of diabetes for each participant, which may have affected our results” (Line 230). The meaning of this sentence is not clear. And, if the course of diabetes is known, what can be done to increase the validity of the results?

Tables: Conventionally, percentages should be calculated as column percentages rather than row percentages. For example, the percentage of the males with diabetes should be 48.3% rather than 14.2% (Table 1). The same applies to Table 2 and Table 5. This will match with the description in the Results. Please revise the tables, as appropriate.

Table 1: Please round the values to sensible precision in reporting. For example, one decimal place is sufficient for height and weight.

Table 3: There is no need to show the prevalence in the table.

Table 3: The columns for odds ratio and P value should be placed in reverse. That is, the column for P value should be placed on the right side of the odds ratio.

Table 3: The variables included in the multiple logistic regression model should be listed in the footnote.

Table 4: Are there notable differences in predictive accuracy of LAP and BMI between males and females?

Table 5: Are there notable differences in the distribution of LAP between males and females?

References: They are a number of errors in the style of references. For example, title case rather than sentence case should be used for journal titles. Please double check according to the journal style.

Reviewer 2 ·

Basic reporting

This is a study of the association between diabetes and lipid accumulation product (LAP) taking advantage of a huge study sample. The nature of the study is confirmatory and replicates earlier studies.
The language is mostly clear, but the text needs rewriting as there are typos like for example “liguid” instead of “lipid” several places. Furthermore, since this is a cross sectional study there is only an association between diabetes and LAP, and it is then not appropritae to describe LAP as a marker of diabetes or stating that is exhibits a higher diagnostic accuracy.
The paper lacks a description of the basic study in which 10% of the population in Urumqi was requited to a physical examination. How were they recruited? Have any analysis been performed to compare the participants to non-participants? Are there any bias or confounders to take into consideration.
Could the authors please provide a reference for the BMI definiotions?

Experimental design

The paper lacks a description of the basic study in which 10% of the population in Urumqi was requited to a physical examination. How were they recruited? Have any analysis been performed to compare the participants to non-participants? Are there any bias or confounders to take into consideration.
With regard to statistics, this seems appropriate, but somewhat insufficient with regard to adust met. I lack a description of what the authors adjust for and a justification for the decisions.

Validity of the findings

The resuls are valid, but the papers lack a justification for replication of the known association between diabetes and LAP. The paper needs to be changed to avoid language implying there are more than an ordinary association between LAP and diabetes.

Reviewer 3 ·

Basic reporting

How to get the cut-off? Please explain in section of method.

Experimental design

no comment

Validity of the findings

1、What is the meanings of calculating the prevalence of
diabetes in LAP and BMI as a quartile? The cut-off value can show the relationship between diabetes and LAP.
2、Is the difference of AUC between LAP and BMI(see tab 4) statistically significant.?please explain

Additional comments

The sampling size is large, but AUC area is less than 0. 75. It is not suitable to be a screening criteria. Should statistical methods be used properly?

---

## Round 0.2 · Minor Revisions

The final decision is still pending some minor changes to be take into consideration.

Reviewer 1 ·

Basic reporting

The authors have adequately revised their manuscript.

Experimental design

The authors have adequately revised their manuscript.

Validity of the findings

The authors have adequately revised their manuscript.

Additional comments

The authors have adequately addressed all my comments except there is a typographical error in both Table 3 and 4. The upper range (55.68) of the 3rd quartile of LAP should not be greater than the lower range (≥ 55.67) of the 4th quartile of LAP. Please fix the error.
In addition, the full form for WC in Table 1 should be given in the footnotes.

Reviewer 3 ·

Basic reporting

no comment

Experimental design

no comment

Validity of the findings

no comment

Additional comments

no comment

---

## Round 0.3 · accepted · Accept

All the reviewers' concerns have been correctly addressed.

Reviewer 1 ·

Basic reporting

The authors have adequately revised their manuscript.

Experimental design

The authors have adequately revised their manuscript.

Validity of the findings

The authors have adequately revised their manuscript.

Additional comments

The authors have adequately addressed my comments.